# Rural Drinking Water Safety under Climate Change: The Importance of Addressing Physical, Social, and Environmental Dimensions

**Jeremy Kohlitz** *⬡, **Joanne Chong and Juliet Willetts**

Institute for Sustainable Futures, University of Technology Sydney, PO Box 123, Broadway, NSW 2007, Australia;
Joanne.Chong@uts.edu.au (J.C.); Juliet.Willetts@uts.edu.au (J.W.)
* Correspondence: Jeremy.Kohlitz@uts.edu.au; Tel.: +61-0451-1826-10

**Abstract:** This paper explores the physical, social, and environmental dimensions of how climate change impacts affect drinking water safety in a rural context in developing countries. Climate impacts, such as contamination or the reduced availability of preferred drinking water sources due to climate-related hazards, threaten water safety in rural areas and these impacts will likely worsen as climate change accelerates. We qualitatively examined these impacts in a community in rural Vanuatu using three approaches side-by-side: adaptation, vulnerability, and resilience. We employed a mixed methods case study methodology that combined semi-structured interviews, technological and environmental surveys, and observations. We demonstrate the influence of physical infrastructure design, social structures mediating water access, and the availability of multiple sustainable water resources on water safety with respect to climate impacts. We also show how the initial problematization of how climate affects water safety can influence subsequent actions to address, or overlook, issues of infrastructure design and maintenance, social equity, and natural resource management for water access. Improvements to rural drinking water safety management in the context of climate change should take a pluralistic approach, informed by different conceptualizations of climate impacts, to account for the varied causal pathways of reduced water safety for different members of a community.

**Keywords:** adaptation; climate change; developing countries; resilience; rural water services; vulnerability; water quality; water, sanitation and hygiene

## 1. Introduction

Climate and water experts are confident that people living in rural areas around the world will experience major impacts from climate change through water supplies [1]. Climate-related events, such as intense rainfall, severe storms, dry spells, extremely hot days, and storm surges, damage or destroy water supply infrastructure, diminish the availability of water resources, and reduce the quality of water used for consumption [2,3]. Human-induced global warming has already led to an increase in the frequency, intensity, and amount of heavy precipitation events; more frequent heatwaves; and mean sea level rise worldwide [4,5]. Climate-related risks for human and natural systems will continue to increase as global warming continues, especially for poor and vulnerable populations [6]. Climate-related water safety risks, which are already important to manage, will likely become increasingly critical to address in rural areas for the foreseeable future.

Strategies for drinking water safety management that address risks related to climate variability and change are needed to deliver safely managed water services in rural areas of the developing world. Safely managed water services are those that are accessible on premises, available when needed,

and free from contamination [7]. The majority of people living in low-income and lower-middle-income countries reside in rural areas [8]. Although data availability is poor, the proportion of the rural population in the least developed countries using improved water supplies that are free from fecal and priority chemical contamination is estimated to be 30% [7]. Climate-related hazards are one driver of the use of contaminated water sources, as we explain later in this paper. Moreover, the poorest and most marginalized people are likely to bear a disproportionate burden of climate change impacts when accessing safe water as a result of their relatively high levels of exposure to hazards and diminished capacity to respond [9]. Drinking water safety management strategies that account for climate-related risks could reduce the proportion of people consuming water from contaminated water supplies.

In this paper, we explore the contributions of three different approaches—adaptation, vulnerability, and resilience—to understanding how climate impacts, such as contamination or the reduced availability of preferred drinking water sources due to climate-related hazards, threaten water safety in a rural developing country context. The purpose of the paper is to present empirical evidence of how each approach frames climate impacts differently and to discuss the implications for policy and practice.

This paper is structured to first cover existing literature on how climate-related hazards (e.g., floods, drought, storms, and sea level rise) compromise water quality and water access in rural developing country contexts and to introduce the theories of the adaptation, vulnerability, and resilience approaches. We then present the methodology and results of a case study in rural Vanuatu which applies these three approaches to reveal a breadth of considerations for climate impacts on water safety. Finally, the implications of the findings for drinking water service policy and practice with respect to climate change are discussed.

## 2. Climate and Water Safety

### 2.1. How Does Climate Variability and Change Influence Drinking Water Safety in Rural Areas of the Developing World?

In this paper, we refer to drinking water safety as the consumption of safe drinking water. Safe drinking water is defined as water that "does not represent any significant risk to health over a lifetime of consumption, including different sensitivities that may occur between life stages" [10]. Climate impacts, through normal climate variability or climate extremes driven by global warming, influence drinking water safety by reducing the quality of preferred drinking water sources and/or causing people to shift to alternative sources that pose health risks. Although climate impacts also influence access to safely managed water services by affecting water reliability, physical accessibility, and affordability, the focus of this study is on water safety.

Table 1 summarizes the ways in which hazards associated with existing climate variability and ongoing climate change can detrimentally affect drinking water safety in relation to rural water supplies in developing countries.

**Table 1.** Climate impacts on water safety in rural contexts.

| Climate-Related Hazard | Impact on Water Safety | Ref. |
|---|---|---|
| Frequent and/or intense rainfall events | Greater trend of *E. Coli* and thermotolerant coliform contamination in boreholes, piped schemes, and rainwater harvesting systems. | [11] |
| | Increased surface runoff that carries fecal matter from soil and latrines into surface and groundwater sources. | [12,13] |
| | Contamination of groundwater sources from sanitation containment units via underground pathways through soil or aquifers. | [14] |
| | Increased agitation of the layer of sludge at the bottom of rainwater harvesting containers which causes pathogens to be suspended from the sludge into the water column. | [15] |

**Table 1.** *Cont.*

| Climate-Related Hazard | Impact on Water Safety | Ref. |
|---|---|---|
| Dry spells or droughts, and extremely hot temperatures | Reduced stream and river flow which raises pollution concentration. | [16] |
| | Increase in growth of algae and some toxic bacteria in surface water and increase in wildfires that raise contaminant loads in surface water, which exert strain on water treatment processes (if present, such as in rural towns). | [17,18] |
| | Diminished groundwater recharge, combined with over-abstraction, leading to salinization of groundwater sources. | [19] |
| | Unavailability of water at improved water sources causing users to access more distant water sources which raises the likelihood of contamination while transporting water home | [20,21] |
| Extreme storms and cyclones | Increased surface runoff that carries fecal matter from soil and latrines into surface and groundwater sources. | [12,13] |
| | Damage or destruction of water treatment facilities or other water supply infrastructure that results in people resorting to unimproved or distant water sources. | [3] |
| Sea level rise | Permanent changes to salinity of groundwater. | [19,22] |
| | Increased likelihood of marine flooding (e.g., from storms surges) and groundwater inundation leading to salinization of wells and boreholes. | [23,24] |
| | Rising groundwater tables which increase the risk of sanitation pollution to groundwater sources. | [14] |

Climate change projections vary across regions of the world, but it is likely that climate change will worsen these impacts on rural water safety in most places. The rising mean sea levels, increasing number of heatwaves, increasing frequency and intensity of heavy rainfall worldwide, and increased risk of drought in some regions [4,5] increases the likelihood and severity of climate-related hazards impacting drinking water safety. Hence it is imperative to consider their management with increased urgency.

*2.2. Three Approaches to Account for Climate Change in Drinking Water Safety Management*

In this section, we briefly present the theoretical basis for and practical examples of three approaches to drinking water safety management drawn from global environmental change and water, sanitation, and hygiene (WASH) literature. These three approaches—namely, adaptation (also known as risk hazard), vulnerability, and social-ecological system resilience—are well-established in broader climate change, disaster, and development discourses and policy [25–29]. For example, an adaptation approach was used in the United Kingdom Climate Impacts Programme to guide national investment decisions on managing expected climate risks, a vulnerability approach was used in national drought management policy in Brazil to reduce unequal suffering, and a resilience approach has been used to analyze the failure of agricultural production in the Goulburn Broken Valley of Australia [26]. It is important to note that these approaches are not mutually exclusive and can be effectively employed together. However, we present them discretely here to illustrate their different potential contributions to drinking water safety management.

The adaptation approach focuses on making adjustments to a system in response to actual or expected climate hazards and their impacts in order to offset potential harm. The approach typically follows the steps of (1) identifying where and when certain climate hazards may appear, (2) assessing the extent to which they can cause losses (e.g., in terms of reduction in water quality), and (3) how the impacts of hazards may be offset or moderated by adaptation actions [30]. In short, the risk that a particular hazard poses for a system is assessed, then adaptations are designed to pre-empt and/or manage that risk.

The risk that a climate hazard poses to a system is a product of the system's exposure and sensitivity to the hazard [31]. Exposure is defined in general as the degree, duration, and/or extent to which a system is in contact with, or subject to, a hazard, while sensitivity is the degree to which a system is modified or affected by a hazard [32,33]. Projections of future climate scenarios are typically used to assess how climate change may increase or decrease levels of system exposure to a particular climate hazard (e.g., a projected increased an intense rainfall events in a region by 2050). The methods for assessing a system's sensitivity to a climate hazard can range from computerized models to simpler dose-response functions (observing the change in effect on a system as levels of exposure to a

hazard change) based on experiences and understanding of the system's behavior [34]. Adaptations to the system that aim to reduce risk are then designed and may be assessed in terms of their cost-effectiveness or other multi-criteria analyses [30].

Water Safety Plans (WSPs) are a WASH analogue of the adaptation approach that focus on physical risks to water quality. WSPs ensure water safety through the assessment and management of risks to water quality from catchment to consumer [35]. In the WSP process, hazards and hazardous events are identified; risks are assessed in terms of the likelihood/frequency that the hazard may occur and the severity/consequences if it does occur; and appropriate control measures for reducing risk are designed, implemented, and monitored. A climate-resilient WSP guide has been developed which follows the same WSP logic but additionally guides users to consider the future and present risks to water safety that climate variability and change create [36].

In public discourse, the term "adaptation" is sometimes used in association with vulnerability or resilience approaches (e.g., adaptations that reduce vulnerability or enhance resilience). There is much conceptual overlap between the approaches that is beyond the scope of this paper to discuss. We refer to adaptations as actions taken to adjust the water system (comprising water infrastructure and technologies, water resources, and their management) to resist specific anticipated hazards.

A vulnerability approach seeks to enable people, particularly the most disadvantaged social groups, to address climate change impacts in general by addressing the root causes of their vulnerability. The vulnerability of people, defined as a propensity or predisposition to be adversely affected by climate change [37], is conceived to be strongly influenced by social and political processes [38]. These socio-political processes unequally expose people to climate hazards, make them more susceptible to harm from climate change, and restrain their ability to influence responses that would benefit them [38,39]. Climate change, in turn, exacerbates poverty and inequality, which deepen vulnerability [40]. This approach raises questions of who is most vulnerable and why and how vulnerability is differentiated [41]. Through the identification of who is most vulnerable to climate change impacts and why, interventions may be designed to address inequalities and empower people to more effectively respond to impacts.

A vulnerability approach to climate impacts on water safety would first seek to understand the inequalities and power relationships within a rural town, community, or household, then assess the extent to which these make people more vulnerable to reduced water safety from climate impacts. However, the WASH sector has engaged little with vulnerability approaches in relation to climate change, instead tending to evoke technological and biophysical-focused responses to climate change [42,43]. Although it does not have explicit consideration of climate change, one example of a vulnerability approach to drinking water safety management is the Equitable Water Safety Planning guide [44]. The Equitable Water Safety Planning guide follows the WSP steps, but instructs users to seek the meaningful participation of different social groups in the WSP process, identify diverse user groups, and investigate their differential experiences with water prior to understanding how they are affected differently by hazards [44]. It also instructs users to implement control measures that address the root causes of differential exposure to hazardous events. A similar process could be followed with a focus on how the diverse experiences of water users create differential exposure to climate-related hazardous events presently and in the future.

Resilience has been conceptualized in numerous ways that focus on different aspects of society or nature (e.g., psychological, disaster, and ecosystem resilience) [45], but the social-ecological system (SES) has emerged as a common analytical focal point in the climate change–development nexus literature [46]. An SES is any system comprising social and ecological components that interact in complex and adaptive ways. SES resilience is defined as "the capacity of a system to absorb disturbance and reorganise so as to retain essentially the same function, structure, and feedbacks—to have the same identity" [47].

An SES resilience (hereafter referred to as "resilience") approach focuses on enhancing the capacity of an SES to adapt to change, particularly unexpected change, through reorganization while ensuring

the functionality of the SES is maintained [47,48]. Reorganizing means that one or more components of the SES (which may be social actors and organizations, physical infrastructure and technologies, or environmental resources) adjust to changes in environmental conditions. Resilience approaches strongly emphasize the role of the natural environment in providing services [49]. Although research is still emerging on the conditions that make a system resilient, researchers have compiled "principles" for building resilience in systems [50], as shown in Table 2.

**Table 2.** Principles for building resilience in systems.

| Principle | Definition |
|---|---|
| Maintain diversity and redundancy | Optimize levels of diversity and redundancy of SES components such that there are multiple options and insurance for responding to disturbances. |
| Manage connectivity | Understand the way and degree to which SES components are connected to one another, and strengthen connections that spread useful material or information while weakening connections that propagate disturbances. |
| Manage slow variables and feedbacks | Identify slow-changing variables that are key to keeping a system stable and prevent the variables from crossing thresholds that would cause system collapse. Strengthen feedback loops that keep key variables within thresholds and weaken feedback loops that do the opposite. |
| Foster complex adaptive systems thinking | Promote a worldview or mental model that views the world as comprising dynamic and interacting systems. |
| Encourage learning | Encourage learning through experimentation and monitoring, especially in real-time. |
| Broaden participation | Actively engage all stakeholders in management and governance processes. |
| Promote polycentric governance systems | Implement multi-scalar, nested, and collaborative governance systems that are matched to the scale of the problem. |

Adapted from [50].

To date, applications of an SES approach to WASH services have primarily been conceptual [51,52], but emerging evidence demonstrates how some resilience principles can support drinking water safety. Access to multiple types of water sources enables people to switch between sources if a climate hazard (e.g., flooding) causes one type of source (e.g., wells) to become contaminated and unusable [53] ("redundancy and diversity" principle). People do not necessarily choose the safest sources amongst their multiple options [54,55], but interventions can provide guidance to water users and service providers about which water sources are most likely to be safest to drink from under different climate conditions [20,56]. In a study in Ethiopia, the real-time monitoring of and response to drought impacts on rural improved groundwater supplies led to increased rates of their functionality so that fewer people needed to rely on emergency water trucking [57], which is considered an unimproved water source and often unregulated in terms of water quality [58] ("encourage learning" principle). Expert opinions suggest that decentralized water supply infrastructure reduces the potential for widespread contamination and public health risk from increasingly frequent and intense rainfall events [59] ("connectivity" principle). However, direct evidence of this in a rural setting is absent.

## 3. Materials and Methods

### 3.1. Case Study Site

Vanuatu is a sovereign archipelago nation located in the South Pacific Ocean approximately 1750 km east of Australia. The total population is estimated to be approximately 276,000, with 75% of people living in rural areas [7]. The country has a tropical climate with distinct dry and wet seasons, although the seasonal cycle can vary considerably based on the phase of the El Niño Southern Oscillation [60]. Average temperatures range from 24 to 27 °C, and average monthly rainfall ranges from approximately 100 mm in September to over 300 mm in March [60].

Recent climate trends and projections of climate change in Vanuatu show increases in temperature, mean sea level rise, and significant uncertainty related to rainfall. Annual and seasonal mean and maximum temperatures in Vanuatu have increased significantly since 1948, and sea level has risen at a higher rate than the global average [60]. Future climate projections over the course of the century suggest an increase in the number of very hot days; more frequent days with extreme rainfall; a continued increase in sea level; less frequent but more intense cyclones; and significant uncertainty as to whether annual and seasonal rainfall will increase, decrease, or stay the same [60].

The 2017–2030 Vanuatu National Water Policy recognizes that water supplies in rural areas are typically managed by community-based committees and calls to "professionalize" management by rewarding committees that develop efficient management systems [61]. The policy also calls for increased access to financing for households to purchase drinking water products, including rainwater storage tanks [61]. However, the extent to which the policy has been operationalized to date is unclear.

The case study site was on a low-lying limestone island located 3.5 km offshore of Lakatoro—the capital of the neighboring Malekula Island. The size of the island is approximately 1.1 km². Seven villages, comprising a combined population of approximately 700, existed on the island. Most community members are fishers, although some grow and sell produce or work in Lakatoro. The villages were treated as a single community in this study because they shared water supplies, were in close proximity to one another, and jointly made decisions on community development issues. The location of the case study site within Vanuatu is marked in Figure 1.

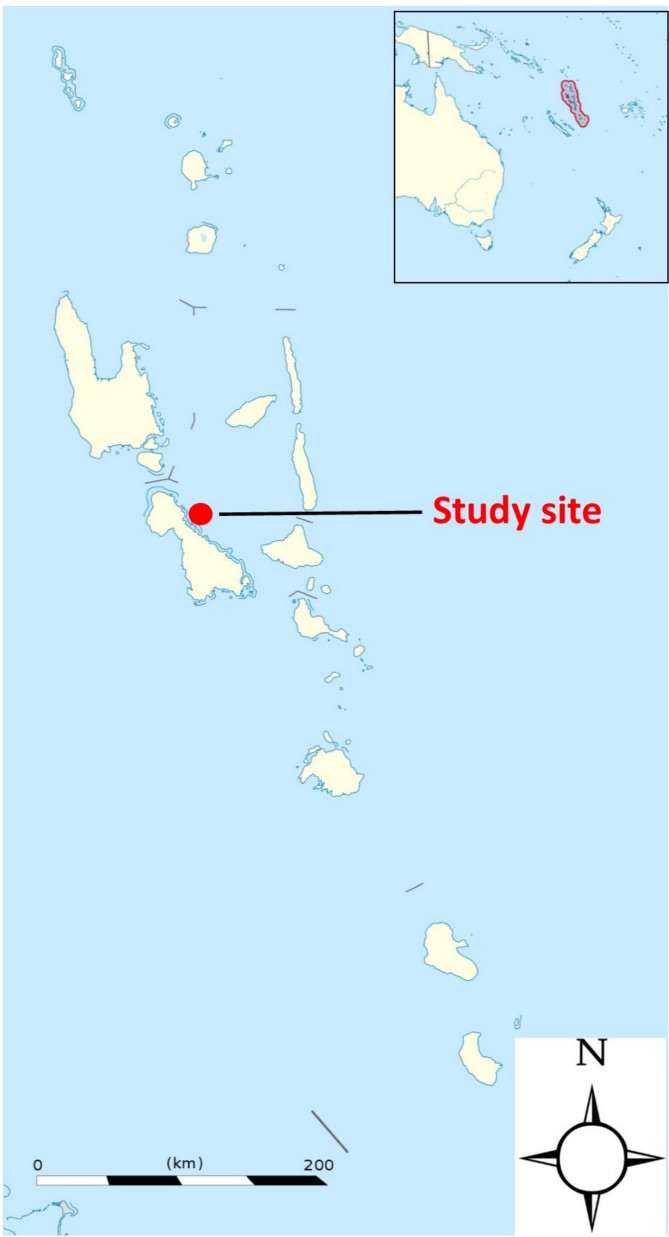

**Figure 1.** Location of the study site community in Vanuatu (Source: modified from http://commons.wikimedia.org).

Water needs on the island are primarily met through the joint use of eight unprotected hand-dug wells and several dozen rainwater harvesting systems spread throughout the community. There are no significant surface water bodies on the island. Only one of the eight wells had a cover (that did not seal). All the well parapets were under 1 m tall, and five well parapets were under 50 cm tall. None of the wells had platforms or lining. Water was retrieved from each well using rope and buckets, except for one well that was fitted with a solar pump. Data on the technical characteristics of the solar pump were not collected. Groundwater on small low-lying islands typically exists as a shallow freshwater lens that sits above seawater [19]. The wells were informally managed collectively by the householders that used them. Figure 2 shows the layout of the community and approximate location of the wells.

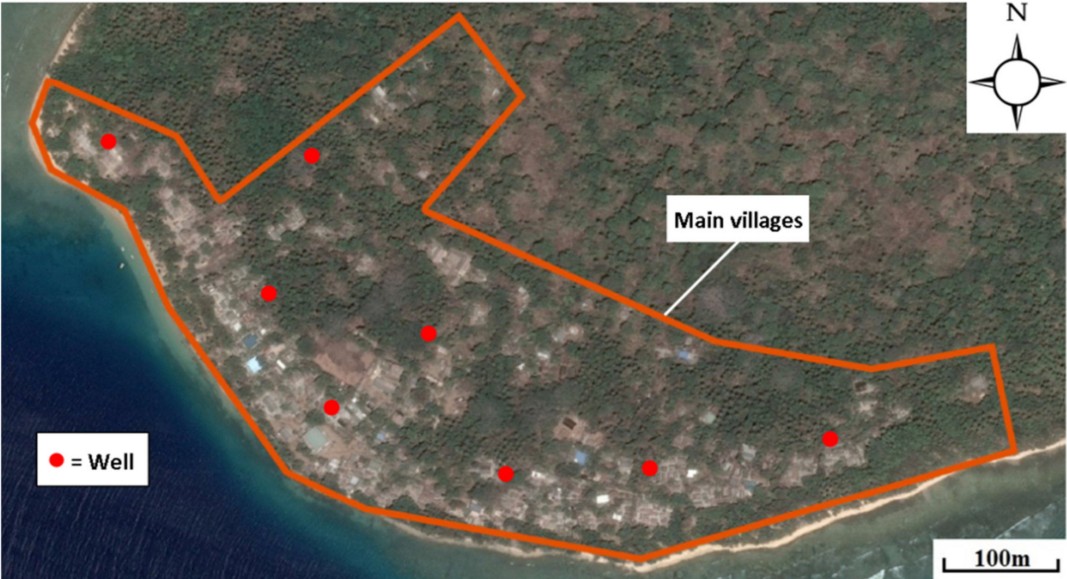

**Figure 2.** Study site community (Source: ©2017 Centre National D'études Spatiales (CNES)/Airbus, DigitalGlobe, Westminster, Colorado, CO, USA).

Most rainwater harvesting systems were attached to domiciles and built by homeowners. Storage receptacles for these systems included pre-fabricated plastic tanks, in situ underground cement tanks, and other locally sourced materials such as drums or barrels. Homeowners managed their own private domestic rainwater harvesting systems. A few communal rainwater harvesting systems were installed on communal buildings like the church, school, and meeting hall by external support agencies. These communal systems have no management entity in place.

*3.2. Methods*

A qualitative case study methodology was followed to demonstrate how climate variability and change impacts can influence drinking water safety in a rural context using the adaptation, vulnerability, and resilience approaches. The adaptation, vulnerability, and resilience approaches were chosen to explore climate impacts on water safety because they are influential on theory, practice, and policy in relation to global environmental change across a range of sectors [25–29]. Because an existing conceptual framework for applying these theories to a rural water service did not exist at the beginning of this study, we inductively used concepts from the adaptation, vulnerability, and resilience theories to construct different framings of climate impacts on water safety. Case studies are commonly used to conduct research or assessments in adaptation, vulnerability, and resilience studies because they provide in-depth detail of the complex ways that climate interacts with society and nature [41,49,62]. Our methods included semi-structured interviews, technological and environmental surveys, and observations of water supplies and how community members interact with them under different weather conditions.

The case study site presented in this paper is a rural community in Vanuatu that accesses water via community-managed systems. This was a suitable site for this study because Vanuatu is classified as having a lower-middle-income economy [63], and community management continues to be the most common water service management model in rural areas of developing countries [2].

The assessments of how climate impacts influence drinking water safety in the community were primarily collected through semi-structured interviews. An interview guide was prepared with questions focused on climate impacts on water management and water access at household and community levels in relation to adaptation, vulnerability, and resilience concepts. In relation to adaptation concepts, interviewees were asked about how frequently they experienced various climate hazards, how different climate hazards affected the water supply functionality and access, and the actions that households and the community commonly took in response to expected or actual climate hazards.

In relation to vulnerability concepts, the interviewees were asked about participation in decision-making about water supplies, household ability to meet water needs, the perceptions of fairness in the community relating to water access, and gender norms around the community and household water management.

In relation to resilience concepts, the interviewees were asked about practices in accessing multiple water sources, backup options when primary water sources failed, how water management practices changed across seasons, innovations in response to climate impacts, and if and how government authorities beyond the community scale provide support to help households meet water needs.

The participants were recruited from different spatial areas within the community to gain a variety of perspectives using different water supplies and were identified via transect walks and snowballing. After obtaining consent from the participants, all the interviews were conducted face-to-face in Bislama. The field team comprised a white male researcher and a Ni-Vanuatu female assistant. The assistant aided in interpreting and the interviews were recorded on a voice recorder. The interviews were conducted at the choice location of the participants, usually the home. A total of 29 interviews were conducted. All the participants were over the age of 18. Due to cultural norms, it was sometimes a challenge to find families comfortable with a female member speaking on their behalf. Consequently, the sample was gender biased, with 17 male participants and 12 female participants.

All the interviews were transcribed into English by the field researcher and analyzed qualitatively. The transcriptions were read multiple times then coded deductively and inductively. Provisional coding [64] was used, whereby a predetermined list of codes derived from adaptation, vulnerability, and resilience theories was used to find empirical evidence of key concepts. Descriptive coding [64] was also used to inductively identify themes that were not captured by the predetermined list. The findings were developed and elaborated on through freestyle writing on the evidence gathered by the codes.

Other methods were used to corroborate the responses of interview participants. The water infrastructure was surveyed using a WSP-style risk assessment [65], whereby potential contamination hazards and risks were identified and documented for community and household water systems. The water catchments were also surveyed to note land use activities and whether the natural vegetation was altered around water sources. Finally, the field researcher was based in the community for one month and recorded the experience through observation notes and journaling. In particular, the researcher observed the conditions of water supplies on dry and rainy days and noted sanitary risks or changes in their functionality. The researcher also observed community members collecting water from communal water points on dry and rainy days and noted how water was collected, by whom, and whether queuing occurred. These observations and assessments were used to gain more insight about the issues reported by the research participants during the interviews.

This mix of methods was used to document cause-and-effect relationships between climate hazards and outcomes in the community, which strengthens the internal validity of the study. Although the impacts discovered in this study may not be generalizable to all rural settings due

to the context-specific nature of climate interactions with society and nature, the lessons learned as elaborated on in Section 5 of this paper are widely applicable to rural water services.

All the participants gave their informed consent for inclusion before they participated in the study. The study was conducted in accordance with the Declaration of Helsinki, and the protocol was approved by the University of Technology Sydney Human Research Ethics Committee (Ref no. 2015000306) and the Vanuatu Cultural Centre.

## 4. Results

In this section, we problematize the impacts from climate variability and climate change for water safety in the participating community through three approaches: adaptation, vulnerability, and resilience. We identify observed, reported, and potential issues for water safety relating to existing climate impacts and how climate change could worsen them.

### 4.1. Adaptation Approach—Increased Physical Risks from Climate Hazards to Water Safety

An adaptation approach can be used to assess the current and future physical risks that climate variability and climate change create for water safety. The risks for the community's water supplies are made with reference to the climate trends and projections for Vanuatu described above.

The wells in the community are exposed to contamination from heavy rainfall. The participants described incidents of surface water runoff carrying debris into wells and degrading the water quality: "*When we sleep during the night, if it rains too much, it will go inside the well. So we must clean the well again*". Technological surveys of the wells found that missing well covers and short parapets could allow the direct ingress of floodwater into the wells. During the community stay, substantial pooling around the wells on rainy days was observed. This could allow for ingress underneath the well parapets, because none of the wells had platforms or lining. Although no latrines were observed within 20 m of the wells, the ground in the island has a shallow layer of soil over porous limestone, which could facilitate the pit latrine pollution of wells from longer distances during wet periods. An increase in days with heavy rainfall, as projected, would be expected to heighten contamination risks to the wells.

The future salinization of the wells due to sea level rise is also a concern. The participants reported that water from one well has always had a salty taste. Another well that had a solar pump installed on it produces a noticeably salty tasting water during dry periods. None of the participants reported that climate-related events affected the salinity of the water in the other wells. However, it was noted from environmental surveys that all the wells were located within a few hundred meters of the coastline. Groundwater lenses on small islands are known to become thinner and more susceptible to saline intrusion during dry periods [66]. Further, it is widely accepted that rising sea levels seriously threaten to make more permanent changes to the salinity of groundwater resources on low-lying islands [19,22]. Increasing sea level rise from climate change, combined with a potential for reduced dry season rainfall (although uncertain), threaten the water quality of the wells.

The rainwater harvesting systems in the community, the only improved water sources, are sensitive to high temperatures and decreased rainfall. The participants reported that rainwater storage tanks commonly became empty during hot and dry periods: "*When the sun is strong for a long time, we use up* [the water in the tank] *I think after one, two, or three months*". Some owners of domestic rainwater harvesting systems reported being able to ration water to last through the dry season, but this is more challenging when the dry season is hotter and drier. When households are unable to access water from rainwater tanks, they usually revert to drinking water from the unimproved hand-dug wells. The risk of water shortages in rainwater harvesting systems, and households consequently drinking from unimproved wells, could increase if the number of very hot days grows as projected, or if the dry season or annual rainfall decreases.

An increase in the number of days of extreme heat could cause some rainwater tanks to crack. Many rainwater tanks in the community are constructed in situ underground by pouring concrete into molds or by binding cement blocks with mortar. Three participants mentioned that their aboveground

cement rainwater tanks were prone to leakage, and it seems likely that underground cement tanks likely leak (or allow underground ingress) as well. High temperatures can adversely affect the mechanical properties and serviceability of setting and hardened concrete [67]. Therefore, an increase in extremely hot days could increase the tendency of cement rainwater tanks to leak and allow ingress from contaminated groundwater.

*4.2. Vulnerability Approach—Unequal Capacity to Respond to Climate Impacts Exposes Some to Water Safety Threats More than Others*

Using a vulnerability lens, the impacts of climate change on water safety in a community can be interpreted in terms of who can most (or least) readily access safe water and why. A salient example of this in the community is the ability to access rainwater harvesting systems—the only improved water source on the island.

Owning a domestic rainwater harvesting system is advantageous in the community because it provides access to an on-site improved water source and an additional layer of water security by complementing the wells. Communal rainwater harvesting systems, such as one built on a church, are free for the whole community to use. However, there is no mechanism for rationing, so they run out of water quickly because families rush to collect as much water from them as they can while water is still available. The wells are less attractive because the well water is sometimes turbid, many community members are aware the well water is unsafe for drinking without treatment, and retrieving water from and queuing at distant wells is wearisome. Consequently, dozens of households in the community mobilize scarce resources to procure domestic rainwater harvesting systems for their families in order to secure safe drinking water.

Although domestic rainwater harvesting systems are popular in the community, the cost of constructing one is prohibitive for many families. Participants remarked that the upfront cost of materials, especially for the tank, prevented them from building a system: "*Sometimes it costs too much. Like the cost of a tank, to make blocks, buy cement*". The wealthiest families, or those with family members sending remittances, purchased pre-fabricated polyethylene tanks which technological surveys showed had the fewest sanitary risks. In situ cement block tanks were a cheaper option but have relatively more sanitary risks. Yet, income-earning opportunities for rural families in Vanuatu are difficult to come by. Consequently, poorer households without a rainwater system often have no choice but to use unprotected wells that are more likely to be contaminated or rely on a neighbor's rainwater system.

Households that rely on a neighbor's domestic rainwater harvesting system are subject to the rules of access that the owner sets for them. For example, the authority of the owner to grant permission to use their system is respected by others. As one participant described: "*If you want to get* [water] *someplace, you must ask. If the owner is not there, you must wait. When he returns, you ask if he can allow it or not*". Another participant conveyed discomfort with using another household's rainwater system: "*Sometimes you feel that you are going too often. You feel ashamed to go*". During interviews, some owners described rules that they set for use of their stored water, such as that water could be collected for drinking only during dry periods. Thus, rainwater system owners were in a position to dictate the terms of water access for non-owners because ownership granted them authority.

Further, as the proportion of households that self-provision water supplies increases, the households that are left behind are increasingly burdened with looking after the wells. During the rainy season when the wells become visibly contaminated or damaged from intense rainfall, village members typically pool resources to repair and restore the wells. With the increasing number of families who meet their water needs during the wet season with rainwater harvesting systems and choose not to contribute to well maintenance during this time, relatively poorer families who still rely on wells are left with a greater share of the maintenance burden.

This is relevant with respect to climate change because, as the adaptation approach section describes, climate change increases risks to water safety especially for the wells. Wealth distribution within the community is a powerful determinant of who will be most exposed to this increased risk.

*4.3. Resilience Approach—Human-Environment Interactions Threaten the Capacity of Water Resources to Deliver Safe Water*

Taking a resilience approach, the threat that climate variability and change pose to water safety in the community can be assessed in terms of the water system's tolerance for variability and volatility and avoiding critical thresholds. In this case, we consider the "water system" to comprise the available water resources, the physical water infrastructure, the water users, and the water management structures. In this section, we present evidence of how selected resilience principles are relevant for water safety and climate change (evidence was not found for all the principles in this study, but this is reflective of resource constraints of the study. Other principles may still be relevant).

The diversity of multiple, decentralized water supplies gives the community a degree of tolerance for the impacts of climate variability on water quality. The proliferation of discrete rainwater harvesting systems in the community, in addition to the eight wells, means that community members have numerous options for accessing water (resilience principle of redundancy). If rainwater systems are dry due to prolonged dry periods, water is still available from the wells, which participants remarked have never gone dry before. If intense rainfall contaminates the wells, water is available in rainwater tanks. Having different water supplies that are not affected the same by various climate hazards makes it less likely that a single climate hazard cuts of all water access (resilience principle of diversity). Furthermore, having numerous discrete water supplies helps to ensure that failures to water supplies are more likely to be isolated and not completely cut off water access (resilience principle of connectivity). This arrangement helps to deal with the future uncertainty in rainfall patterns due to climate change.

A critical threshold for the water service in the community is the level of groundwater that can be abstracted before the salinization of a water source occurs. On small islands, groundwater pumping technology risks the salinization of the freshwater lens through "upconing", whereby brackish water in the "transition zone" (the zone where the freshwater lens and seawater meet underground) is pulled upward [19]. The participants remarked that this had been experienced at one well where a solar pump was installed to make collecting water less arduous: "*Now it is becoming salty. I think because the dry season is too strong, it is causing a lack of water. The solar pump is pulling it but it is becoming salty*". Once the water in the well became saline, participants reported that they began rejecting it for drinking, which marks the important threshold. Although upconing is reversible by discontinuing or slowing pumping rates, repeated incidences of upconing can cause the thickness of the transition zone to permanently increase [68]. This means that the salinization threshold can potentially move to become more restrictive if upconing repeatedly occurs. The management of this feedback (e.g., by limiting the amount of abstraction by season) is critical for ensuring the well is not permanently salinized (resilience principle of management of feedbacks). At the time of the visit, the participants remarked this was not being proactively managed.

The salinization threshold is relevant to climate change impacts because of mean sea level rise. Climate change-driven sea level rise could combine with the increased water abstraction from wells to cause the salinization threshold to become more quickly and frequently crossed, resulting in degraded water quality and potentially the abandonment of the wells.

Another example of a feedback mechanism relates to how land management affects the water quality of the wells. Vegetation and good soil structure promote the infiltration of rainwater into the ground [69], which is then purified as it percolates toward the water table [70]. An environmental survey found that the development of living spaces on the island has led to the clearing of natural vegetation and compacting of the topsoil. As a result, the land's infiltration capacity has decreased and surface runoff has increased, which causes problems with spills of contaminated runoff into open wells. The participants reported that this, in turn, leads users to take up coping responses such as treating the well water through boiling (often using firewood sourced by clearing more vegetation, which continues a feedback loop).

This feedback mechanism that affects water safety may be exacerbated by climate change. If the number of days with heavy rainfall increases as projected, this may lead to increased contamination events from surface runoff and an increase in trees being cut down to source firewood for boiling water as a treatment.

## 5. Discussion

In this section, we discuss the potential responses that could be taken to address the existing and possible climate impacts on water safety in the community and the implications of our findings for WASH policy and practice.

### 5.1. Possible Responses to Climate Impacts on Water Safety through Each Approach

One outcome of problematizing climate variability and change impacts on water safety through different approaches is that it produces sharply different recommendations for how to respond. Table 3 summarizes the potential interactions between climate and water safety presented in this paper and lists the possible responses that government or civil society organizations (CSOs) could make to improve or maintain water safety.

**Table 3.** Summary of potential interactions between climate and water safety and the possible responses government or civil society could make.

| Approach | Climate Interactions with Water Safety | Possible Responses |
|---|---|---|
| Adaptation | Contamination of wells due to increase in number of days with heavy rainfall | Install well platforms, high parapets, and well covers. |
| | Salinization of wells due to sea level rise | Dig new wells further inland. Increase number of rainwater systems or install desalinization units. |
| | Water shortages and leakage of rainwater tanks due increase in number of extremely hot days | Increase rainwater storage capacity and improve tank building techniques and materials. |
| Vulnerability | Poorer households cannot afford domestic rainwater harvesting systems that are more likely to provide safe water than dug wells | Subsidize the cost of rainwater tanks for poorer households. |
| | Owners of private domestic rainwater harvesting systems in part dictate terms of water access for non-owners | Establish a management entity for communal rainwater tanks that prioritizes water needs of poorest households. |
| | Poorer households sharing a greater burden of well maintenance in the wet season | Upgrade wells and allocate resources to support their operation and maintenance. |
| Resilience | A variety of discrete water supplies gives community members multiple options for safely accessing water | Upgrade and support upkeep and utilization of water supplies drawing on different water resources. |
| | Increased groundwater abstraction threatens to salinize wells | Support community to set rules to moderate groundwater abstractions and monitor well salinity. |
| | Land clearance contributing to increased surface runoff and contamination of groundwater | Restore vegetation on the island. Support alternative treatment methods to boiling to break feedback loop. |

It should be noted that it is assumed that the responses in Table 3 would improve water safety in this context based on commonly accepted knowledge about contamination pathways, but further research is needed to verify this and to understand who would benefit most. Future quantitative studies may examine the association between water quality, seasonal or climatic events, and technical or managerial characteristics of the water supply and catchment system. For example, associations between the frequency of intense rainfall and high sea levels with groundwater contamination, or groundwater monitoring to model water balance across seasons. Mixed methods research could provide insights on how financial (e.g., rainwater tank subsidies) or institutional interventions (e.g., rule-setting or management support) influence the decisions of different people to drink safe or unsafe water when certain climatic conditions are experienced. Evidence generated by studies like these would support informed decisions about developing actions to address climate impacts on water safety.

The impacts and responses in Table 3 are only a small sample of those that could be generated using each approach to examine climate change impacts on rural water safety, but illustrate the influence that initial conceptualizations have on actions taken to respond to climate change. Responses following an adaptation approach primarily involve investing in making technological improvements in order to reduce risks to water quality. The vulnerability approach leads to responses that focus on water

governance and management that supports equality. Finally, responses following the resilience approach are both biophysical and social, but tend to relate to water resource management.

Each set of responses are mostly complementary and can and should be employed together to ensure people drink safe water as climate hazards worsen. Systematic climate hazard identification, risk assessments, and control measure implementation can reduce the water quality risks from climate change in the community. However, the community capacity to implement control measures is not homogenous and initiatives are needed to ensure the outcomes are equitable. For example, increasing rainwater storage capacity could help to address water shortages, but additional attention is needed to ensure that poorer households can afford rainwater tanks or get water when they need it from communal tanks. Ensuring equitable access to a water supply with climate risk management measures in place can also be complemented by the resilience responses that aim to manage social-ecological interactions to support the community to sustain multiple water resources for them to flexibly draw on as the climate changes in unpredictable ways.

If each of the different approaches are not accounted for in government or CSO interventions to improve water safety, it is easy to see how water safety can be inadvertently undermined in other ways. For example, sealing the wells and adding pumps may help to prevent contamination from surface runoff, but also risks salinizing the groundwater in the future if sea level and abstraction rates rise. Increasing equitable access to diverse water supplies can enable people to dynamically choose which water source to drink from depending on the prevailing seasonal or climate conditions, but if they are not trained on how to recognize and manage sanitary risks, they may choose to drink from less safe sources (as documented by [54]). A wide perspective of water safety that encompasses the physical, social, and environmental dimensions is needed to ensure that people can always drink safe water under climate change.

*5.2. Implications for Policy and Practice*

The different ways that climate change can influence drinking water safety in rural settings calls for pluralism in policy and practice. Policy should recognize the contributions of both natural and social sciences in understanding how people can be exposed to unsafe drinking water due to present climate variability and future climate change. Case studies such as this that illustrate this or research that directly shows evidence of climate impacts on water quality and public health outcomes help to inform policy. Frameworks that link different disciplinary knowledge bases can help to bring different scholarly contributions on climate change impacts on water access together in a coherent way [71]. There is also a need to translate research findings and conceptual thinking such as that provided in this paper into concrete, practical interventions that can be taken to scale.

Partnerships between WASH researchers and implementers, and climate change authorities, will be important for developing practical responses to climate impacts on water safety. Much of the latest thinking on responding to climate change is complex and overwhelming to WASH stakeholders that are new to this field. Researchers and climate change authorities that have worked in other sectors that have successful implemented climate change programs can help government and WASH CSO implementers to make sense of various concepts and how they might apply to water safety. Implementers can provide practical experience of water safety actions that can be feasibly implemented in rural settings and considerations for taking those actions to scale. Collaborative workshops between CSOs, other in-country stakeholders, and researchers to brainstorm and critique ideas and engage with other sectors that have implemented successful climate change programs to learn from their experiences could help with developing practical solutions. The production of empirical evidence through case studies or experimental methods is needed to assess the effectiveness and equity of intervention outcomes.

While from a theoretical perspective, water safety management interventions should address all of these multiple dimensions, it should be recognized that resources for investment in a rural developing context are scarce. For example, in a given situation funding may be needed to upgrade

water infrastructure to better manage risks, diversify water sources, and offer subsidies to poorest households to increase household safe storage capacity, but budgets may be limited to carry out all of these. Decisions on which investments are most likely to most effectively improve water safety and public health outcomes must be evaluated on a case-by-case basis. The involvement of different community social groups, governments, and CSOs in discussions, as well as consultations with climate experts, should inform these investment decisions.

## 6. Conclusions

Climate change threatens drinking water safety in rural areas in multi-dimensional ways. In applying adaptation, vulnerability, and resilience analyses side-by-side in a case study in Vanuatu, we have demonstrated the contributions that different approaches can make to problematizing climate variability and change for rural drinking water safety in developing countries. The adaptation approach raises the importance of recognizing water quality risks created by present and future climate hazards. The vulnerability approach points to how social processes cause climate impacts to create more water safety issues for some than others. The resilience approach highlights the value of enabling people to have flexibility in how they access safe water under unpredictable climate conditions, and the need to sustain water resources. These complementary approaches can and should be employed together when considering drinking water safety management under climate change. If any approach is applied in isolation, it is possible that important dimensions of water safety will be overlooked or undermined.

A key implication of this research is that the global WASH sector's focus to date on technological and infrastructural improvements for managing risks from climate hazards has made important contributions to water safety but must be expanded. Climate change impacts will not affect water safety for people living in rural areas to the same level, even for people living in the same community. Further, not all climate hazards and their risks for water safety are predictable, and some will emerge over many years as climate change continues to accelerate. A broader appreciation of the multivariate threats to water safety that climate variability and change pose in the present and future and corresponding actions to address those threats are essential to achieving universal access to safe water in an ever-dynamic world.

**Author Contributions:** Conceptualization, J.K.; methodology, J.K.; validation, J.C. and J.W.; formal analysis, J.K.; investigation, J.K.; data curation, J.K.; writing—original draft preparation, J.K.; writing—review and editing, J.C. and J.W.; supervision, J.C. and J.W.; project administration, J.K., J.C. and J.W.; All authors have read and agreed to the published version of the manuscript.

**Funding:** This research received no external funding.

**Acknowledgments:** We thank Tiffany Vira for assisting in the fieldwork component of this study. We also thank the community for welcoming us and participating in this study.

**Conflicts of Interest:** The authors declare no conflict of interest.

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
