# Peer review of "Rural Drinking Water Safety under Climate Change: The Importance of Addressing Physical, Social, and Environmental Dimensions"

_resources, doi:10.3390/resources9060077_

Round 1
Reviewer 1 Report
The topic is very interesting and challenging , basically the linkage between the climate change from one hand and the socio-economic and environmental aspects are things significantly needed in the future. the Author/s providing a very good information with scientific sounds.however I believe the main major area that need to be revisit and organize is the structure of the work ,below are also my comments wish to find it helpful to the Author/s to improve this nice piece of work:
- Abstract
Lines 15 and 16 repeated sentence
- Introduction
Lines between 51-60 sounds more like methodology and has nothing to do in the introduction sector, please focus on the objective here and embedded this paragraph in the Method section
Lines 61-68 need to restructure to highlight and reflect only the main objective of this work.
The introduction is too long with narrative details that can be turned to figures, a good example the section 1.1 the way describing the climate change how influencing the drinking water safety in particular for rural areas, too many basic details could be easily illustrated by one figure to demonstrate the domino affect. Please reflect the text to figure that will
The 1.2 section is completely related to methodology should be integrated there and has nothing to do with introduction.
- Materials and Methods
Lines 219-223 repeated paragraph recommend to remove.
Recommend to start the 2.1 with the case study and region or interest then start 2.2 with the methods used (considering the above comments mentioned where most of paragraphs in introduction has included the methods or work and integrating this paragraph with the lines from 223 to 278 to form the methods part.
The author/s didn’t justify why they using these 3 approaches? What is the distinguished by using these approaches? Could sentences of justification is highly recommended.
Fig.1 need to provide with better quality , the resolution is not comparative and please add the north sign to the figure for better orientation.
Since no surface water is available and the island relay on Groundwater and rainwater ,the followings information are needed : 1. The basic metrological information about the region of interest (Temperature,perscipitation,humidety..etc.) 2. The basic information for the 8 hand-dug water wells depth, yield and elevation if available. 3. The water table level and major type of aquifer. 4. The percentage of contribution of rainwater and the groundwater for householders consumption and for which purposes (I.e. agriculture, domestic ..etc.)
Who is the authority (if any) responsible on gives the digging water wells or establishing a harvesting unite? and what is the policy of water rights (if any)?
What is the specification of the solar pumps that have been used?(type of panel, size of battery, weight of energy..etc.) Please add couple sentence in this context.
- Results
Lines 329-331 , very good point, and thats why we should add more details about the wells depth and water table as highlighted above, the frequency of this type of flash rain happened also would help to figure out the weight of potential contamination and impact.
Lines 333 - 340 here there are some information and about the wells ,I would highly recommend to allocated these information and the information needed that highlighted previously to the case study site paragraph.
Lines 344-370 , sea water interaction with groundwater is a common thing at this case and this is why need to mentioned about the type of aquifer available,however a big question to rise here in terms of differentiate between the contamination from the rainwater and the sea water interaction with groundwater, this I would say out of the scope of this current project but could be a good idea to raise this question in this work to build a future project by means of standard experimental field work fishing water samples for rainwater and groundwater with/without the sea water interaction to conduct a conceptual model for the sources of water contamenation, this will be linked to the climate change and sea level increasing.
Lines 358-360, recommendation for future work to have a numerical model for the water balance verses the climate change impact to predict the shortages, such a model would be helpful to give indicators and actions how to bridge the lean time.
Author Response
Thank you to Reviewer 1 for your constructive comments and suggestions which have considerably strengthened our paper. Please see our responses in the PDF attachment.

Reviewer 2 Report
I revised the paper titled “Rural drinking water safety under climate change: The importance of addressing physical, social and environmental dimensions”. I think that the matter of the paper is in the scope of the Journal and I found it very interesting. The authors use three different approaches to evaluate how the climate change affect the drinking water safety in rural areas. The implementation of the adaptation, vulnerability and resilience approach in combination with the environmental surveys and observations illustrate the current situation and practices in the drinking water safety management in the study area and how the upcoming climate change can affect the water safety. Also, the proposed responses is a significant part in this study because the authors propose possible solutions for the improvement of drinking water safety.
My suggestions are the following:
Page 1 – line 25: Some keywords such as adaptation approach, vulnerability approach and resilience approach might be added
Page 1 & 2 – line 28-68: The paragraphs should be aligned
Page 2 – line 54: “drawn from the global environmental change literature”. Some references might be added and example of scientific outcomes if possible
Page 2 – line 74-76: “Climate impacts…….pose health risks”. A reference might be added.
Page 3 – line 126: “[26,27,28,29,30]” should be [26-30]
Page 3 – line 142-145: “Methods for……….of the system’s behaviour”. References might be added.
Page 5 – line 200-201: A blank line should be added.
Page 7 – line 295: “km2” should be “km2”
Page 12 – line 476-477: A blank line should be added.
Page 12-14 – Discussion: The summary of the possible responses that a government or a civil could take, is quite useful since through these actions could be achieved the improvements of drinking water safety and the reduction of the adverse impact of climate change. However, it would be wise tο be added examples of research studies in which similar possible responses have been implemented and the outcomes they had.
Page 15-18 – References: Some references have to be written in accordance with the Instructions for Authors of the journal
Author Response
Thank you to the Reviewer for your constructive feedback and suggestions which have considerably strengthened our paper. Our responses to each comment are in the attached PDF.

Round 2
Reviewer 1 Report
The Author/s did a significant adjustment and answered most the questions and comments. the work has been improved, however four answers still did not highlighted/reflected on V.2 please review the four comments below and reflect your answer on the revised version. for better orientation I have order them as (V.1 Q mean the reviewer question in the version 1) Answer= your previouse answer on V.1 Q. and the new comment= the reviewer latest comment.
Good luck
V.1 Q: What is the specification of the solar pumps that have been used?(type of panel, size of battery, weight of energy..etc.) Please add couple sentence in this context.
Answer: We agree this information could be useful for understanding the over-abstraction/salinization issue experienced in the community, however data on the technical characteristics of the solar pump were not collected.
New comment: Then please add this to your work “data on the technical characteristics of the solar pump were not collected”.
V.1 Q: Lines 329-331 , very good point, and thats why we should add more details about the wells depth and water table as highlighted above, the frequency of this type of flash rain happened also would help to figure out the weight of potential contamination and impact.
Answer: We agree that additional data on the groundwater, and on frequency and intensity of rainfall events, could be used to inform a model of water contamination. Although it was not within the scope of this study to collect this level of data, we have added lines under section 4.1 to note that this type of data is useful for informing climate response actions.
New comment: I couldn’t find this added line you mentioned in 4.1 or may be you didn’t highlighted as the others comments. If not then please add.
V.1 Q: Lines 344-370 , sea water interaction with groundwater is a common thing at this case and this is why need to mentioned about the type of aquifer available,however a big question to rise here in terms of differentiate between the contamination from the rainwater and the sea water interaction with groundwater, this I would say out of the scope of this current project but could be a good idea to raise this question in this work to build a future project by means of standard experimental field work fishing water samples for rainwater and groundwater with/without the sea water interaction to conduct a conceptual model for the sources of water contamenation, this will be linked to the climate change
and sea level increasing.
Answer: These are good points in terms of informing some adaptation actions. We have added lines under section 4.1 that more data could be collected to inform responses as mentioned previously.
New comment: Same, I couldn’t find this added line you mentioned in 4.1 or may be you didn’t highlighted as the others comments. If not then please add.
V.1 Q: Lines 358-360, recommendation for future work to have a numerical model for the water balance verses the climate change impact to predict the shortages, such a model would be helpful to give
indicators and actions how to bridge the lean time.
Answer: As above, we have added lines under section 4.1 to note that further data could be collected to inform responses.
New comment: Same, I couldn’t find this added line you mentioned in 4.1 or may be you didn’t highlighted as the others comments. If not then please add.
Author Response
We thank the Review for providing these follow-up comments. We have revised the manuscript and have articulated our responses in the attachment.

Reviewer 2 Report
I have to mention that the authors have made many changes that have covered the majority of reviewers' comments. The overall quality of presentation, the significance of content and scientific soundness of this manuscript have improved significantly. Some minor editorial remarks are only needed.
Page 6 - line 195-214: The references [64] and [65] are mentioned after the reference [60] and before the reference [61]. Therefore, you might check the references again or change their reference numbers in order to avoid misinterpretations.
Author Response
We thank the Reviewer for looking at our manuscript again and identifying corrections. We have revised the manuscript and articulated our response in the attachment.
